# SIMPEL: using stable isotopes to elucidate dynamics of context specific metabolism
Shrikaar Kambhampati [1,7,8] ✉, Allen H. Hubbard[1,8], Somnath Koley [1,8], Javier D. Gomez [2], Frédéric Marsolais [3,4], Bradley S. Evans[1], Jamey D. Young [2,5] & Doug K. Allen [1,6] ✉

The capacity to leverage high resolution mass spectrometry (HRMS) with transient isotope labeling experiments is an untapped opportunity to derive insights on context-specific metabolism, that is difficult to assess quantitatively. Tools are needed to comprehensively mine isotopologue information in an automated, high-throughput way without errors. We describe a tool, Stable Isotope-assisted Metabolomics for Pathway Elucidation (*SIMPEL*), to simplify analysis and interpretation of isotope-enriched HRMS datasets. The efficacy of *SIMPEL* is demonstrated through examples of central carbon and lipid metabolism. In the first description, a dual-isotope labeling experiment is paired with *SIMPEL* and isotopically nonstationary metabolic flux analysis (INST-MFA) to resolve fluxes in central metabolism that would be otherwise challenging to quantify. In the second example, *SIMPEL* was paired with HRMS-based lipidomics data to describe lipid metabolism based on a single labeling experiment. Available as an R package, *SIMPEL* extends metabolomics analyses to include isotopologue signatures necessary to quantify metabolic flux.

Cellular metabolism is defined by fluxes through biochemical pathways that can be quantified with isotope tracers. Bond rearrangements quantified through a single [13]C tracer provide considerable information for hetero- or mixotrophic plant systems with steady state isotopic labeling[1–8] and, when transient labeling is considered, provide insights on autotrophic[9–17] or heterotrophic metabolism[18–20]. Using targeted approaches, gas chromatography mass spectrometry (GC-MS)[21,22], and liquid chromatography tandem mass spectrometry (LC-MS/MS) can provide rigorous quantification of compounds and their isotopologues[23–25] to analyze metabolic network function and may consider spatial attributes through imaging-based MS techniques[26–29]. The attention given to developing different methods reflects the complexity of metabolism that is difficult to assess and can benefit from more than one labeling experiment with different [13]C substrates[4,6–8,19,30–34] or [2]H, [15]N or [18]O-based labeling to resolve pathway use[35–43].

Technological advances now allow monitoring and quantification of large numbers of metabolites through untargeted approaches[44–46] and can link with multi-omics platforms to assess the operational state of a metabolic network, but these new measurement capabilities present several challenges

and directions[47–53] that could partially be addressed through incorporation of tracer studies. Though the acquisition of additional data is not a bottleneck for such studies, manual curation and integration of isotopologues is time consuming and not always compatible with the extensive number of replicates needed for statistically meaningful results. Further, isotopologue monitoring can result in overlapping signals. For example, dual-labeled tracers (e.g., [13]C[15]N-glutamine) produce changes in mass to charge ratio (i.e., *m/z*) in labeled metabolites that indicate the addition of a single neutron from either heavy element and is nominally the same for heavy isotopes from different elements (Fig. 1a). Thus, the capacity to resolve additional fluxes from combining information from multiple heavy isotopes within the same experiment is limited when nominal *m/z* values are recorded.

High resolution mass spectrometry (HRMS) can distinguish metabolites (and isotopologues of heteroatoms) through accurate measurement of *m/z* values that differ by <0.01 Da (Fig. 1a) providing a largely untapped framework to extend isotopic studies and metabolic flux analyses[38,54]. The inclusion of a second heavy isotope in the experimental design increases mass balancing constraints and improves the sensitivity of flux estimates beyond the enhanced data richness associated with additional isotopologue

[1]Donald Danforth Plant Science Center, St. Louis, MO 63132, USA. [2]Chemical and Biomolecular Engineering, Vanderbilt University, Nashville, TN 37235, USA. [3]London Research and Development Center, London, ON N5V 4T3, Canada. [4]Department of Biology, University of Western Ontario, London, ON N6A 5B7, Canada. [5]Department of Molecular Physiology and Biophysics, Vanderbilt University, Nashville, TN 37235, USA. [6]Agricultural Research Service, US Department of Agriculture, St. Louis, MO 63132, USA. [7]Present address: Jack H. Skirball Center for Chemical Biology and Proteomics, The Salk Institute for Biological Studies, La Jolla, CA 92037, USA. [8]These authors contributed equally: Shrikaar Kambhampati, Allen H. Hubbard, Somnath Koley. ✉e-mail: skambhampati@salk.edu; doug.allen@ars.usda.gov

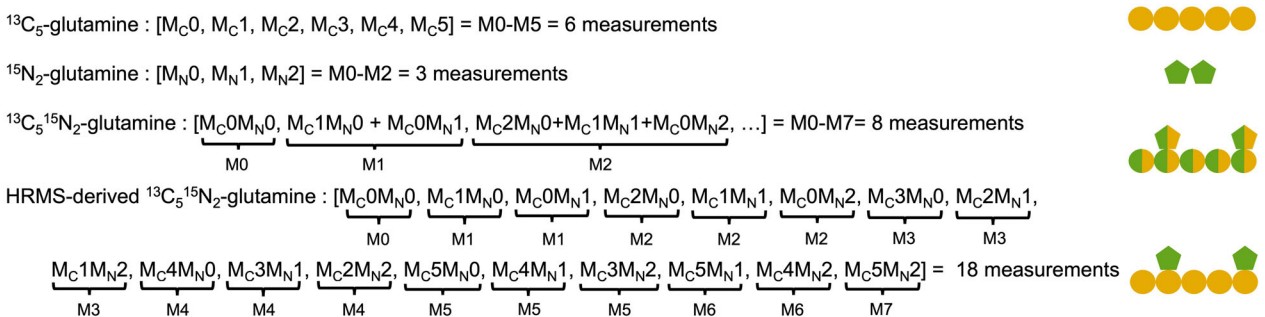

measurements. HRMS instruments scan and quantify thousands of $m/z$ values per second and can easily distinguish isotopologues and compositional changes in mass, for example, the nominal change in $m/z$ of two Daltons that results from either inclusion of two carbon-13 atoms or alternatively a bond saturation or other chemical modifications[38,54]. Because HRMS can measure $m/z$ values quickly, many metabolites actively incorporating isotopes can be identified through an unsupervised, unbiased approach with untargeted data acquisition that is blind to the metabolic network. The enumerating and quantification of metabolomic features detected in samples enriched with multiple stable isotopes remains an analytical bottleneck[55,56], as the number of multivariate isotopologues[35] of a given compound ($I$) increases exponentially with the number of labeled atoms ($N$) for each element ($e$) of the molecular formula, following Eq. (1).

$$I = \prod(N_e + 1) \quad (1)$$

**Fig. 1 | Demonstration of the use of high-resolution mass spectrometry (HRMS) to distinguish dual labeled isotopologues and establish fluxes with confidence.** **a** HRMS has the capacity to resolve dual labeled isotopologues that are indistinguishable using nominal resolution MS, a schematic example with expected separation is shown. M0, M1, M2,….Mn represent the isotopologues that differ by a unit mass, resulting from labeling. **b** Ten-day old Arabidopsis seedlings, transferred to medium contained [$^{13}C_5$,$^{15}N_2$] glutamine and collected after time course labeling, were used for HRMS data acquisition. Acquired data were first pre-processed using XCMS, and post-processing data analysis was performed using SIMPEL. Data tables exported from SIMPEL were used for INST-MFA with INCA. **c**. Network representing the biochemical reactions modeled with INCA to estimate fluxes. Flux values (nmol g$^{-1}$ h$^{-1}$) for each reaction are shown next to the arrows along with CIs in parentheses. Color coding is as follows: yellow represents fluxes established from a single $^{13}C$ experiment, green reflects a map with parallel $^{13}C$ and $^{15}N$ experiments, and the orange fluxes resulted from a dual labeled $^{13}C^{15}N$ experiment. Measurements for isotope enrichment within metabolites for all three models were obtained from the same [$^{13}C_5$,$^{15}N_2$] glutamine labeling experiment, by using different combinations of isotopologues that represent a single ($^{13}C$), multiple parallel ($^{13}C$ and $^{15}N$) or dual

labeled ($^{13}C^{15}N$) experiments (Supplementary data 1, table 1). **d** As an example of the comparative information for modeling, glutamine that has 5 carbons and 2 nitrogen atoms is provided. In the $^{13}C$ stand-alone experiment there are 6 isotopologue measurements (M0-M5) (used for modeling the single $^{13}C$ labeling experiment). For the $^{15}N$ stand-alone experiment there are 3 isotopologue measurements (M0-M2). Thus, as two separate experiments there are 9 measurements (used for modeling the parallel labeling experiment). When a single $^{13}C^{15}N$ experiment is made, a nominal mass instrument would not be able to distinguish whether the increase in m/z was due to N or C, thus only the additional m/z values (i.e., M0-M7 isotopologues, a total of 8 measurements) were modeled. Complete HRMS analysis provides fine detail and distinguishes each elemental isotopologue such that 18 measurements can be made and used in principle (data from these measurements were summed to obtain the $^{13}C^{15}N$ isotopologues (M0-M7), the $^{13}C$ isotopologues (M0-M5), and the $^{15}N$ isotopologues (M0-M2) from a single dual labeling experiment). In general, the number of measurements will be dependent on the number of experiments and tracers, however, for the HRMS case the 18 measurements from soft ESI-MS without fragmentation to the backbone can be generalized using Eq. (1). Abbreviations are listed within Supplementary Data 1, Supplementary Tables 2 and 3.

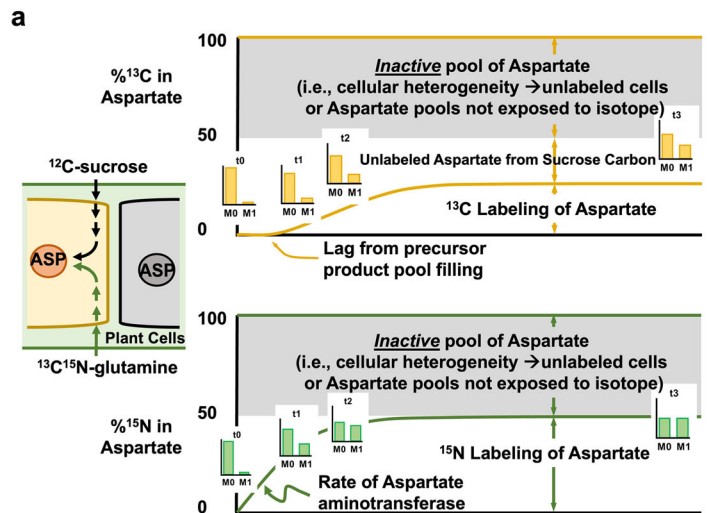
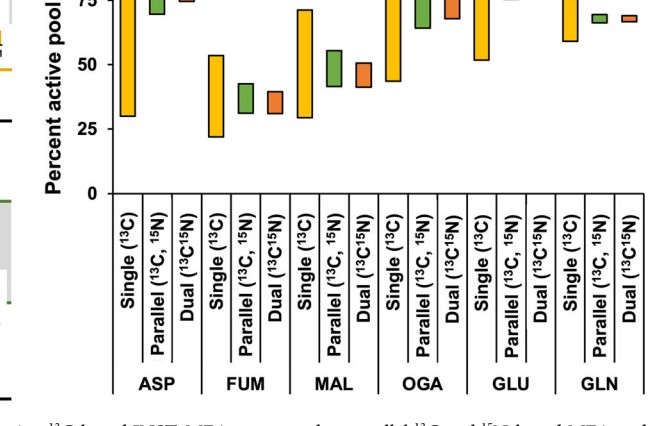

**Fig. 2 | Comparison of active vs inactive pool determination between single, parallel and dual-labeled MFA. a** Hypothetical inactive pool description for aspartate that is a consequence of unlabeled sucrose in the medium and labeling from $^{13}C^{15}N$ glutamine. **b** 95% confidence intervals (CIs) for active pool estimates

using $^{13}C$-based INST-MFA, compared to parallel $^{13}C$ and $^{15}N$-based MFA and a dual labeled $^{13}C^{15}N$-based MFA for key metabolites in the model, plotted as a range. Abbreviations are listed within Supplementary Data 1, Supplementary Table 3.

Computational tools compatible with HRMS isotopologue datasets[57–64] offer technical advances in workflows, address changes in instrumental capabilities and technology, and have started to confront bottlenecks in the analyses of isotope enriched HRMS data and their biological implications. Here a new tool is described, **S**table **I**sotope-assisted **M**etabolomics for **P**athway **EL**ucidation (*SIMPEL*) that capitalizes on HRMS data collected from transient stable isotope labeling experiments. *SIMPEL* automates post-processing data analyses of isotope enriched metabolomics datasets, generates isotopologue distributions, performs natural abundance corrections along with some global analyses to streamline biological interpretations. Two case studies are presented, where the utility of SIMPEL is highlighted to understand central carbon and lipid metabolism. In the first description, a dual-isotope labeling experiment processed with *SIMPEL* was used for isotopically nonstationary metabolic flux analysis, to resolve challenging fluxes that traditionally require data from multiple labeling experiments. Providing network-constraining information by tracking two elements in a single labeling experiment enabled more precise flux estimates. In the second example, HRMS-based lipidomics data were processed using *SIMPEL* to describe dynamic lipid metabolism. Following metabolite precursor-product relationships, the role of PC and DAG in acyl editing was considered within oilseeds.

## Results and discussion

*SIMPEL* was designed as a data analysis tool to compare labeling and exact mass information enabling determination of the elemental composition of isotopically enriched compounds and application of correlation-based approaches to cluster metabolite peaks with similar patterns of isotopic enrichment from transient labeling experiments. *SIMPEL* eliminates challenges in data analysis by providing a platform to compatibly link untargeted metabolomics from HRMS with isotopologue labeling from multi-tracer transient labeling experiments. The software combines pre-processed metabolomics feature-lists from contemporary untargeted metabolomics data analysis software such as XCMS[65] or MZmine2[66] with comprehensive lists of compounds of interest including molecular formulae and retention time to search for and identify target metabolites and their isotopologues. Post-processing data-analyses that can be performed with *SIMPEL* include:

1. Enumeration of isotopologues for a given chemical formula through *m/z* tolerances,
2. Identification of isotopologues from libraries of chemical formulae and retention times,
3. Calculation of isotopologue distribution and average labeling per compound from intensities,
4. Correction for natural abundance (NA) with IsoCorrectoR[67],

**Fig. 3 | $^{13}$C labeled lipidomic data extraction, plotting, and analyses from pre-processed data using SIMPEL. a** Isotopologue distribution plot, where the abundances of each of the isotopologues as a proportion of the sum were plotted on the y-axis and the duration of $^{13}$C labeling (in hours) is shown on x-axis. Both the uncorrected and natural abundance corrected plots generated with *SIMPEL* are shown for PC(36:4) as an example (Supplementary Data 1, Supplementary Table 5). **b** Average labeling calculated using the isotopologue distributions and nmol of $^{13}$C product formed, the latter of which incorporates accurate quantification data for lipids (Supplementary Data 1, Supplementary Table 7). **c.** *k*-means clustering plot with clusters represented as nmol of $^{13}$C product formed on the y-axis (lines represent the mean enrichment and shaded regions represent standard deviation) and duration of $^{13}$C label incorporation on the x-axis. A 2D representation (1$^{st}$ and 2$^{nd}$ principal components) of the same clusters with time collapsed to highlight the compounds present in each cluster is shown on the right (data presented in Supplementary Data 1, Supplementary Table 7).

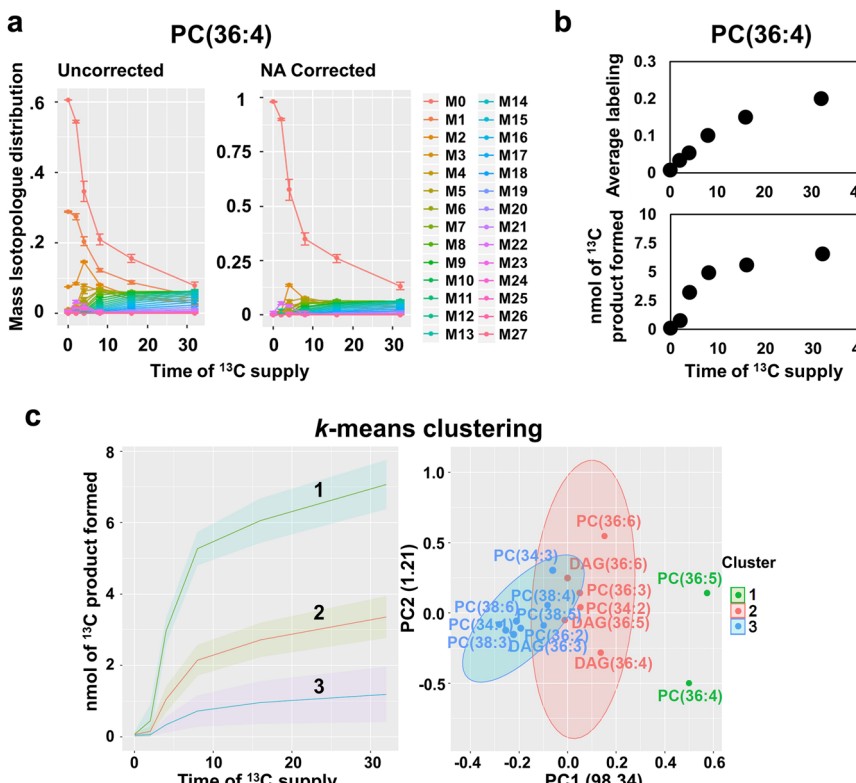

5. Visualization and export of isotopologue distributions and average labeling for all compounds,
6. Global analyses such as principal component analysis (PCA), hierarchical or k-means clustering of compounds based on label enrichment over time.

Two case studies are provided involving *SIMPEL*. In the first investigation, ten-day-old *Arabidopsis* roots were grown on a predefined carbon and nitrogen rich medium[68] supplemented with $[^{13}C_5{}^{15}N_2]$ glutamine (Fig. 1a) and sampled over time (0–8 h) prior to *SIMPEL*-based data extraction and metabolic flux analysis. An untargeted metabolomics dataset generated by a qExactive Orbitrap MS and pre-processed with XCMS was imported into SIMPEL (Fig. 1b) resulting in a post-processed data table containing isotopologue distributions for metabolic intermediates that were used to perform isotopically non-stationary metabolic flux analysis (INST-MFA) with INCA[69] (Fig. 1c). Metabolic networks that included either $^{13}$C or $^{13}$C$^{15}$N atom transitions were modeled with labeling information from a single $^{13}$C experiment, parallel $^{13}$C and $^{15}$N experiments, or with the addition of $^{13}$C$^{15}$N isotopologues that represent information obtained from a single dual-labeling experiment (Fig. 1d). Without HRMS, three independent tracer experiments would be required to obtain these datasets (i.e., one with $^{13}$C, one with $^{15}$N, and one with combined $^{13}$C/$^{15}$N labeling). However, all three datasets could be derived from a single $^{13}$C$^{15}$N-glutamine labeling experiment by summing isotopologue abundances contained in the comprehensive HRMS dataset (Fig. 1d). As noted in Fig. 1d, combining the three nominal mass resolution $^{13}$C$^{15}$N datasets provides fewer measurements than modeling the complete HRMS isotoplogue distribution, but INCA does not currently have the capability to model HRMS data directly.

The inclusion of a second isotope (i.e., $^{15}$N) enabled active metabolite pool assignments and identification of differences that were a consequence of active pools that incorporated unlabeled carbon from sucrose, versus pools that were inactive or not turned over during the time frame of the experiment (Fig. 2a). The inability to distinguish these descriptions normally confounds labeling interpretations and subsequent flux analyses[70].

Here, MFA based on parallel or dual labeling with $^{15}$N reduced inactive pool estimates and recast the associated confidence intervals (Fig. 2b). The reactions that use nitrogen differ from those that exclusively use carbon (e.g., aminotransferase reactions relative to carbon shuttling through central metabolism for organic building blocks), thus, pairing isotopes from multiple elements that trace different aspects of metabolism and achieve isotopic equilibrium in metabolites at different rates can help define more realistic metabolic networks. The confidence intervals for the active pool estimates were significantly reduced for the model including dual labeling compared to parallel $^{13}$C and $^{15}$N labeling experiments (Fig. 2b), likely due to the additional paired $^{13}$C$^{15}$N isotopologues that describe concomitant use of labeled precursors within a metabolite pool.

Compared to conventional flux studies involving multiple parallel labeling experiments[6,7,9,33], a single, dual-labeled experiment can potentially provide additional information[71] from the heavy isotopologue connections that can be quantified through HRMS with *SIMPEL* resulting in more sensitively determined fluxes and smaller confidence intervals (CIs) (Fig. 1c). Fluxes through organelle transport steps that do not involve metabolite interconversions (e.g., OAA transport between mitochondria and cytosol) are difficult to determine with certainty but could be determined by the software through adjacent reactions and thus can benefit from a second elemental isotope signature. Nitrogen label transfer in addition to the $^{13}$C movement during the conversion of OAA to aspartate enables a better determination of the OAA transfer flux. Bifurcated pathways such as glutamate dehydrogenase (GDH) or glutamine oxoglutarate aminotransferase (GOGAT) could also be resolved based on the nitrogen transfer through coordination with aspartate aminotransferase (ASPAT). While two parallel labeling experiments with $^{13}$C-glutamine and $^{15}$N-glutamine, separately, resolved active pool sizes and reliably determined transport fluxes with improved precision compared to a single $^{13}$C experiment, the dual-labeled substrate in combination with HRMS and *SIMPEL* established tighter CIs for all fluxes likely due to the provision of additional labeling information tracking two elements that were complementary in constraining the flux solution. On average, the $^{13}$C and $^{15}$N parallel labeling-

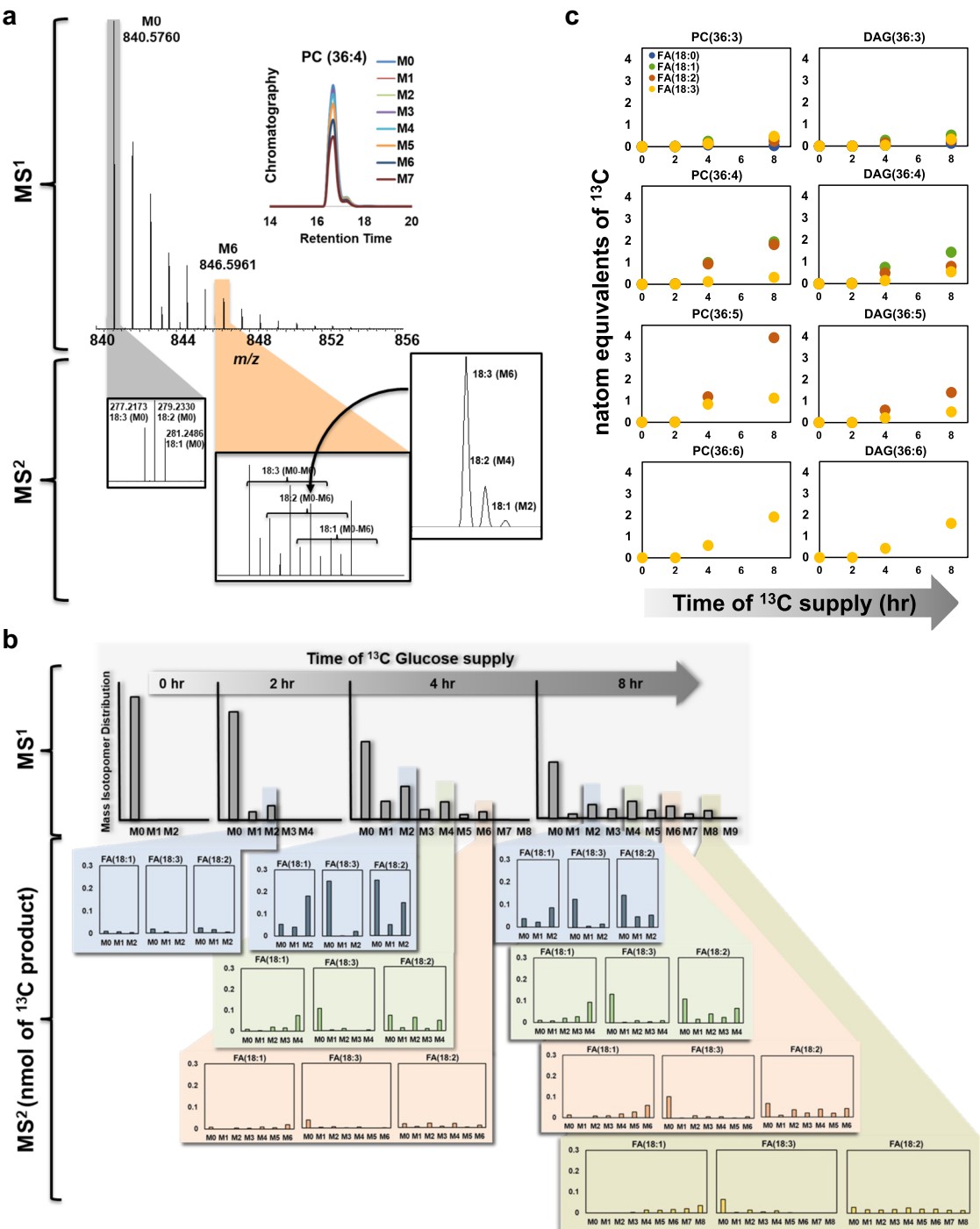

**Fig. 4 | Determination of acyl-chain labeling within lipid molecular species using MS² at high resolution. a** Tandem MS spectra of PC(36:4) and its isotopologues that co-elute in chromatography. HRMS resolves the different isotopologues, highlighted in gray, M0 (840.5760), and orange, M6 (846.5961), in the MS¹ scan. MS² spectra for the unlabeled isotopologue shows 18:1, 18:2 and 18:3 FAs, while the M6 isotopologue shows combinations of labeled FAs that can further be resolved using HRMS. **b** Relative intensities of the isotopologues of PC(36:4) were plotted for each time point (0, 2, 4 and 8 h of ¹³C glucose labeling) using the MS¹ data analyzed by *SIMPEL*.

MS2 spectra for each of the isotopologues were used to determine the distribution of FA labeling within the acyl chains of PC(36:4). Data at MS² represents isotopologue distribution that is adjusted for pool size and the proportions of FAs that constitute PC(36:4) (see methods, Supplemental File 2). **c** ¹³C enrichment in acyl chains of PC(36:x) and DAG(36:x) represented as natom equivalents of ¹³C (see methods, Supplementary Data 1, Supplementary Tables 8 and 9 for detailed description and calculation) using MS² data obtained upon fragmentation of isotopologues.

based MFA and ¹³C¹⁵N dual labeled MFA reduced the CIs by 71 and 84%, respectively, compared to the single ¹³C-based MFA (Supplementary Data 1, Supplementary Table 3).

In a second study, *SIMPEL* was used to probe stable isotope labeled lipidomics data obtained with HRMS. HRMS distinguished the

incorporation of two ¹³C molecules ($\Delta m/z = 2.00671$) from a degree of fatty acid unsaturation ($\Delta m/z = 2.01565$) based on $m/z$ difference of 0.009 Da[38]. Developing *Camelina sativa* seeds, which represent stages corresponding to oil biosynthesis and accumulation, were labeled with [U-¹³C₆] glucose in a time-course experiment (0–32 h). Embryos were excised from pods, and

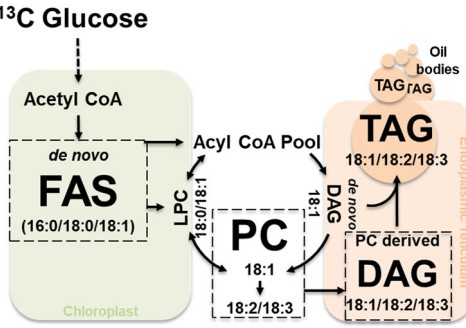

**Fig. 5 | Simplified description of seed oil biosynthesis and the central role played by PC.** Seed oil biosynthesis based on [13]C glucose labeling involves the movement of acyl chains onto and off PC for desaturation, known as acyl editing[72–74] and the PC may additionally serve as the shuttling mechanism for the acyl chain export from the chloroplast to the ER[38,75,79]. FAS Fatty acid synthesis, LPC Lyso phosphatidylcholine, PC Phosphatidylcholine, DAG Diacylglycerol, TAG Triacylglycerol.

cotyledons were separated and assessed to gain insights into fatty acid biosynthesis and lipid metabolic pathways in plants. Untargeted lipidomic datasets generated using a Thermo Fusion Lumos Tribrid MS were pre-processed using XCMS and automatically analyzed by *SIMPEL* (Fig. 3a) resulting in average labeling descriptions for diacylglycerol (DAG) and phosphatidylcholine (PC) that are central to triacylglycerol assembly and polyunsaturation in plants which utilize acyl editing mechanisms[72–74]. Much of what is known about plant lipid biosynthesis has been and continues to be established through inspection of [14]C labeled lipids[72,75–77]; however, HRMS and SIMPEL provide a complementary technique to elucidate the precursor-product relationships and rates of labeling of pathway inter-mediates in living systems when stable isotope investigations are appropriate.

To enable direct comparison of [13]C enrichment between different lipid species, pool size (nmol) was accounted for (see Supplementary Data 1, Supplementary Tables 4–7) to obtain the quantity of [13]C labeled molecules formed[16,78] (Fig. 3b). A *k*-means clustering analysis of the quantities of [13]C PC and DAG species formed over the time course of labeling revealed three distinct clusters (Fig. 3c). Cluster 1 consisted only of two PCs and was the fastest labeled group, whereas clusters 2 and 3 contained a mixture of other PC and DAG molecular species (Fig. 2c). The results, obtained from a single untargeted assessment of [13]C-labeled lipids, confirmed previous [14]C labeling results that PC is the most rapidly labeled lipid in seed oil production[72,75]. Further, the identification of rapidly labeled PC(36:4) and PC(36:5) species suggested the role of these lipids in connecting fatty acid biosynthesis in the chloroplast with lipid assembly in the endoplasmic reticulum. Membrane lipids such as PC may be a conduit for the direct shuttling of acyl chains from the chloroplast to the ER[38,75,79] through contact sites or to extend channeling mechanisms[80–83]. PC comprises a significant percentage of the lipids present at the chloroplast envelope[84] at around 40%[75]. Tandem MS analysis of [13]C enrichment within the fatty acids of PC and DAG molecular species at 2-, 4- and 8-hour time points (Fig. 4a, b, Supplementary Data 1, Supplementary Tables 8 and 9) indicated highest labeling in 18:1 followed by 18:2 acyl chains at the earliest time point, which represented most of the label enrichment (Fig. 4c). By 4 h, acyl chains within PC(36:5), i.e., FA(18:2) and FA(18:3), were enriched, consistent with desaturation over time. DAGs presented a similar trend, but were delayed relative to PC, with FA(18:1) of DAG(36:4) being the most labeled at both 2 and 4 h while FA(18:2) and FA(18:3) accumulated higher [13]C enrichments by 8 h. The comparable FA enrichment in PC and DAG molecular ions provided a semi-quantitative assessment of the important role of enzymes that interconvert DAG and PC through head group (choline) exchange such as PC:DAG choline phos-photransferase (PDCT)[85] and CDP-choline:DAG choline phospho-transferase (DAG-CPT)[86]. Our contemporary understanding of seed oil

biosynthesis involves de novo synthesis of saturated or monounsaturated fatty acids in the chloroplast, which are then converted to acyl-CoAs that are incorporated into PC for desaturation with acyl chains either released as part of acyl editing, passed to DAG through lipid headgroup exchange (Fig. 5), or used as the third acyl chain in TAG biosynthesis by phospholipid: dia-cylglycerol acyltransferase (PDAT)[87]. As PC and lysoPC (LPC) can be found on the surface of the outer chloroplast membrane[75,84], and a significant percentage (30% or more) of lyso-phosphatidylcholine acyltransferase, LPCAT, activity is associated with chloroplasts[88], the labeled FA(18:1) and FA(18:2) of PC is consistent with PC serving as a carrier of acyl chains that are channeled to the ER[38,75].

Further, the analyses revealed significant labeling in 16:3/18:3 mono-galactosyldiacylglycerol (MGDG) (Supplementary Fig. S1) that suggest chloroplast Kennedy pathway flux of 16:3 in plants not generally considered in green seeds of oil producing crops. Such examples highlight the oppor-tunity for untargeted metabolomics to elucidate undescribed aspects of metabolism with stable isotope labeling and HRMS.

## Conclusions

*SIMPEL* provides high throughput identification and quantification of isotopologues from untargeted HRMS datasets arising from transient iso-tope labeling studies, presenting an opportunity to discover and describe novel metabolic activities. With dual labels ([13]C, [15]N), *SIMPEL* enabled more precise flux estimations of central metabolism from a single experiment. Similarly, stable isotope labeling of an oilseed indicated PC and DAG roles in acyl editing that were consistent with channeling of acyl chains through PC to the ER, and MGDG labeling that support the presence of an active chloroplast Kennedy pathway in a green oilseed. *SIMPEL* is available as an R package for comparative analysis of any pulse labeled stable isotope enrichment datasets generated using traditional metabolomic data acquisition.

## Methods
### Plant growth and experimental conditions

For the metabolomics study using dual-isotope labeling, wildtype *Arabi-dopsis* ecotype Columbia seeds were grown on vertical plates at 22 °C under continuous light (ca. 70 μmol m$^{-2}$ s$^{-1}$), on a defined nutrient medium previously described[68] for metabolomics with dual-isotopic labeling. The medium consisted of 10 mM potassium phosphate (pH 6.5), 5 mM KNO$_3$, 2 mM MgSO$_4$, 1 mM CaCl$_2$, 0.1 mM FeNaEDTA, micronutrients (50 mM H$_3$BO$_3$, 12 μM MnSO$_4$, 1 mM ZnCl$_2$, 1 mM CuSO$_4$ and 0.2 mM Na$_2$MoO$_4$), 1% sucrose and 1% agar. Ten-day old seedlings were transferred to plates containing the same medium, except the nitrogen source was replaced with 10 mM [[13]C$_5$,[15]N$_2$]glutamine. Root tissue was excised after exposure to medium containing labeled glutamine for 2, 4, 6 and 8 h to represent time course incorporation of carbon and nitrogen into metabo-lism. Untreated roots were used as unlabeled (0 h) controls. Each plate yielded ~100 mg of root tissue and served as a single replicate. Four repli-cates per sample type were collected and flash frozen using liquid N$_2$ for total metabolite extraction.

Stable isotope labeled lipidomics was performed with developing embryos of *Camelina sativa*. Plants were grown in greenhouses with day/night temperature maintained at 22/20 °C, 40–50% relative humidity, and 16 h day/8 h night photoperiod. Intact siliques during the seed filling growth stage (15 days after fertilization) were excised and placed in sterile media containing a modified Linsmaier and Skoog medium[89,90] with Gamborg's vitamins (Sigma) and 5 mM MES buffer adjusted to pH 5.8. Fifty mM [U-[13]C$_6$]glucose was used as labeled substrate[91], and the composition of the remaining carbon and nitrogen sources represented maternal phloem composition to minimize metabolic perturbation and to maintain pseudo in vivo conditions. Silique culturing was performed in a 96-well plate with 0.3 mL of medium and a single silique per well, under continuous light (250 μmol m$^{-2}$ s$^{-1}$)[9]. Tissue was collected and flash frozen immediately after each time point (2, 4, 8, 16 and 32 h). Uncultured siliques excised from the maternal plant were used as unlabeled (0 h) controls. Frozen tissue was

sectioned, on top of dry ice, to excise embryo from the siliques and to separate cotyledons from the embryo axis. Cotyledon samples were extracted and analyzed for lipids in triplicates.

## Total metabolite extraction

Frozen *Arabidopsis* root tissue was homogenized using a tissue lyser, and extraction was carried out using 1 mL of 4:1 methanol: water (v/v) with incubation in an ultra-sonication bath for 30 min followed by shaking for 30 min at 4 °C. The mixture was then centrifuged at $21,000 \times g$ for 10 min at 4 °C; supernatant was transferred into fresh tubes and evaporated to dryness using a speedvac centrifuge at ambient temperature. Dried residue was re-suspended in 200 µL of 1:1 methanol: water (v/v), filtered using 0.2 µm PTFE micro centrifuge filters and transferred to glass vials for HILIC-HRMS runs.

Frozen cotyledon samples from *Camelina* were homogenized using a tissue lyser and the extraction of lipids was carried out using a phase separation method previously described[92]. Briefly, 1 mL 7:3 methanol:-chloroform (−20 °C) containing the ultimateSPLASH™ ONE lipid mix (Avanti Polar lipids, Alabaster, AL) as internal standard (1:20 dilution) was added to the samples, vortexed vigorously and incubated on a rotary shaker for 2 h at 4 °C. After incubation, 500 µL of ice-cold water was added to the samples, vortexed and centrifuged at 14,000 rpm at 4 °C for 10 min to achieve phase separation. The upper aqueous phase was carefully removed, 200 µL of methanol was added to the remaining organic phase containing lipids and centrifuged at 14,000 rpm for 5 min to pellet the debris. The organic phase (supernatant) was transferred to a glass tube and dried using a speedvac centrifuge. Samples were re-suspended in 200 µL of 49:49:2 acetonitrile: methanol: chloroform, filtered using 0.2 µm PTFE microcentrifuge filters and transferred to a glass vial for RPLC-HRMS analysis.

## Metabolomics data acquisition using HILIC-HRMS

Chromatographic separation using HILIC was achieved using an Agilent 1290 Infinity II UHPLC system equipped with a SeQuant® ZIC®-HILIC (100 ×2.1 ×3.5 µm) column (EMD Millipore, Burlington, MA). Mobile phases A and B were comprised of 5 mM ammonium acetate (pH 4.0) in water and 90% acetonitrile with 0.1% acetic acid, respectively. A flow rate of 0.3 mL min$^{-1}$ was used to elute compounds with the following gradient: 87% B for 5 minutes, decreased to 55% B over the next 8 min and held for 2.5 min before returning to 87% and equilibrating the column for 3 min. The heated electrospray ionization (HESI) conditions used were as follows; spray voltage, 3.9 kV (ESI+), 3.5 kV (ESI−); capillary temperature, 250 °C; probe heater temperature, 450 °C; sheath gas, 30 arbitrary units; auxiliary gas, 8 arbitrary units; and S-Lens RF level, 60%. Full MS data were collected using a Q-Exactive Quadrupole Orbitrap mass spectrometer (Thermo Fisher Scientific) in both positive and negative ionization mode separately from mass ranges 75–1100 *m/z* and 65–900 *m/z*, respectively, at 140,000 resolution. The automatic gain control (AGC) was set to $3 \times 10^6$ and maximum injection time (IT) used was 524 ms. Top 12 data dependent MS/MS (ddMS2) spectra were also collected for a representative pool of unlabeled samples with MS1 and MS2 data acquired at 35,000 and 17,500 resolution, respectively. The AGC target and maximum IT in the ddMS2 experiment were set to $1 \times 10^6$, 128 ms for MS1 spectra and $1 \times 10^6$, 64 ms for the MS2 spectra. A 2.0 Da isolation window and normalized collision energy of 25 were used for ddMS2 with the underfill ratio set to 1.0% along with 10 sec dynamic exclusion to reduce redundant spectra.

## Lipidomics data acquisition using RPLC-HRMS

Separations for lipidomics were achieved using the loading pump of a Dionex UltiMate 3000 RSLCnano system (Thermo Fisher Scientific) operating at a flow rate of 40 µL min$^{-1}$ equipped with a custom-made C8 column (100 ×0.5 ×5 µm) from Higgins Analytical Inc. (Mountain view, CA) re-packed from a nucleodur C8 Gravity column (Macherey-Nagel, Allentown, PA). Mobile phases comprised of 1% 1 M ammonium acetate, 0.1% acetic acid in water (A) and 1% 1 M ammonium acetate, 0.1% acetic acid in 7:3 (v/v) acetonitrile: isopropanol (B). The following gradient

modified from a previously described method[93] to adapt to micro flow was used; 0-1 min at 55% B, 4 min at 75% B, 12 min at 89% B, 15 min at 99% B, 18 min at 99% B and 20 min at 55% B followed by equilibration up to 30 min. The eluent was sprayed on to the HESI source of an Orbitrap Fusion Lumos Tribrid MS, operated with sheath gas, 25 arbitrary units; auxiliary gas, 5 arbitrary units; ion transfer tube temperature, 300 °C; vaporizer temperature, 100 °C; and S-lens RF level, 60. The spray voltage was 4 kV in both positive and negative modes. Full MS data were collected for mass ranges 450–1200 *m/z* at 240,000 resolution from both positive and negative modes simultaneously, using polarity switch. The AGC target was set to "Standard" and the maximum IT was set to 100 ms.

MS/MS data were collected for representative pooled unlabeled and labeled samples using AcquireX Exclusion-Inclusion lipid characterization template with iterative injections. MS1 data within this experiment were obtained at 120,000 resolution from a mass range of 450–1200 with AGC target set to "standard" and maximum IT set to "auto". MS$^2$ data were also collected at 120,000 resolution within the orbitrap detector, to achieve enough mass resolution for distinguishing $^{13}C$ labeled isotopologues from acyl chain desaturation. A guide to determining the required resolution to sufficiently distinguish m/z of interest is provided within supplementary information (Supplementary text T1). The collision energy mode was set to "assisted" with 25, 30 and 35% collision energies with an activation time of 20 ms. The AGC target and maximum IT for MS2 experiments were also set to "standard" and "auto", respectively. A 0.5 Da isolation window was used and a cycle time of 1.5 s was used within the data dependent mode. The intensity threshold filter, set to $5 \times 10^4$, and apex detection were enabled to prioritize ion isolation at their maximum intensity. Data dependent acquisition was also enabled to obtain MS2 on the most intense ions if target ions were not detected within a given scan.

## Data pre-processing and creation of compound database

Full MS data files in Thermo .RAW format obtained in profile mode were first centroided by conversion to mzML format using the MSConvert application within ProteoWizard[94] with peak picking filter applied. For data collected with polarity switch, the positive and negative mode data were separated using the subset filter within MSConvert. Files were imported into R using the XCMS package[65], and features were detected using the centWave[95] method and 5 ppm mass tolerance. Pre-filter was set to 6 scans with a minimum intensity of 5000; signal to noise ratio and the threshold for noise were set to 5 and $1 \times 10^6$, respectively. Retention time alignment was performed using the obiwarp[96] method, grouping included features present in at least 25% of all samples, allowable retention time deviation was 10 s, and *m/z* width was set to 0.01. The "fillPeaks" function was used with default settings, and data was not imputed. All features identified were exported to Excel for use as an input data table for *SIMPEL*. MS/MS data collected from representative unlabeled samples were used within a commercial software, Compound Discoverer 3.0 (Thermo Scientific), following the "MaxID" workflow. Only the compounds manually verified to have a positive match with the mzCloud MS/MS and/or LIPID MAPS database and a valid chromatographic peak were used to build custom compound databases, for both HILIC and lipidomic methods, which contained chemical formulae and retention times to be used as an input for *SIMPEL*.

## Post-processing of stable isotope enriched data using *SIMPEL*

The XCMS pre-processed data tables and the compound databases (described above) for both dual-isotope labeled metabolomics and single-isotope labeled lipidomics datasets were packaged within *SIMPEL* as example data for users. All the isotopologues and their *m/z* values for compounds within the database were matched against retention time and identified from the pre-processed data using the function "get_table_objects_NA_corrected". The function also includes user defined tolerances for m/z and rt values, which may not be universally applied for all isotope enriched metabolomics datasets, an iterative approach to determine optimal settings may be necessary. The same function also performs NA correction, calculates isotopologue distributions and average labeling for

both NA corrected and uncorrected data and stores them as data objects that are used for plotting based on user preferences. A tutorial describing the usage of each of the functions within *SIMPEL*, and equations that *SIMPEL* uses for calculations, are provided as Supplementary File 3.

### Isotopomer network and determination of fluxes

A reaction network describing glycolysis, TCA cycle, and related amino acid biosynthesis was constructed using a subset of a previously published *Arabidopsis* model[11] and by consulting biochemical literature. The model was adjusted to include carbon and nitrogen atom transitions for the dual-isotope labeling experiment. A list of reactions, and abbreviations can be found in supplementary data 1, table 3. Metabolic fluxes and confidence intervals were estimated using INST-MFA through the MATLAB-based Isotopomer Network Compartmental Analysis (INCA) package, (Version 2.0)[97]. Best-fit values for fluxes were obtained by repeating the evaluation a minimum of 1000 times from random initial values and subjecting the optimized values to a $\chi^2$ goodness of fit test. 95%-confidence intervals were computed for all estimated parameters by evaluating the sensitivity of the sum-of-squared residuals to parameter variations[98]. The MATLAB model files for the three experimental designs described in the results are also provided as Supplementary Data 2, 3 and 4 for convenience.

### Accurate quantification and label enrichment of lipids and their acyl chains

Unlabeled control samples of *Camelina* cotyledons, that serves as controls, were spiked with nonendogenous, deuterium labeled internal standard mix, i.e. ultimateSPLASH™ ONE and used for accurate quantification of PC and DAG lipid molecular species. Both PC and DAG standards contained a d5 labeled C17 chain in the *sn*−1 position and varying chain lengths from C14 to C22 in the *sn*−2 position. Each standard within the mix was considered a "bucket", and the concentration of each analyte was calculated using multiple buckets based on its acyl chain composition as determined using MS[2]. For example, PC(34:x) that contains a C16 and a C18 acyl chain was quantified once using d5PC(17:0/16:1) as bucket 1 and again using d5PC(17:0/18:1) as bucket 2. A mean value of the calculated concentration obtained from all possible buckets was treated as the accurate quantity of analyte. Steady-state metabolism was assumed with an expectation that the pool sizes of lipid intermediates such as PCs and DAGs do not change throughout the course of the labeling experiment. PC and DAG quantities obtained in nmol amounts were used to convert NA corrected isotopologue distributions generated for the pulse labeled samples, through *SIMPEL*, into nmol abundances of isotopologues. A sum of all the labeled isotopologues (M1-Mn) in nmol represented the amount of $^{13}C$ product formed over the course of the labeling experiment. These data were used to for *k*-means clustering analysis to enable comparison of label enrichment between different pools of PCs and DAGs that are otherwise incomparable due to their differences in pool sizes[30]. MS2 data collected from the labeled samples were used to manually assess and integrate the acyl chain composition and labeling within each of the PC and DAG molecular species. IsoCorrectoR[67] was used to correct for natural abundance, and the corrected isotopologue distributions were converted to natom equivalents of $^{13}C$, modified and adopted from Arrivault et al.[78], using the pool size information.

### $^{13}C$ label quantification and data interpretation of acyl chains within Lipids using MS[2] fragments

$^{13}C$ Glucose labeled cotyledons collected from 2, 4 and 8 h time points along with unlabeled seeds that served as controls were run on a Thermo Orbitrap Fusion Lumos tribrid MS through an untargeted exclusion-inclusion lipid characterization template with iterative injections as described in the methods section and referred to by the manufacturer as AcquireX. The MS[2] ion fragments containing acyl chains and their labeled isotopologues were used to calculate the enrichment in the fatty acids of the lipid molecular species. As an example, negative ion mode fragmentation of *m/z* of 846.5961 that is the formate adduct of the M6 isotopologue of PC(36:4) with (18:1/18:3) and (18:2/18:2) acyl chains contains M0-M6

of 18:1, 18:2 and 18:3 fatty acids (Fig. 4a). The M6 weighting is conserved through summed pairs (i.e., M0/M6, M1/M5, M2/M4, M3/M3, M4/M2, M5/M1, M6/M0) of labeling in acyl chains. Intensities of MS[2] acyl chain labeling distributions were averaged for four scans per precursor *m/z* for M2, M4, M6 and M8 isotopologues. Labeling distributions for each labeling time (2, 4 and 8 h) were corrected for NA, using IsoCorrectoR, and converted into nmols of $^{13}C$ isotopologue by accounting for pool size of the lipid and its isomeric proportions (Fig. 4b) (see Supplementary Data 1, Supplementary Tables 8 and 9 for calculations). As a proxy for labeling in lipids, even labeled isotopologues offset by factors of two (M0, M2, M4, M6 and M8) were integrated and analyzed at each time point representing the total labeling within lipid coming from a uniformly labeled source of glucose. The $^{13}C$ enrichment within acyl chains of each lipid were determined by weighting labeled acyl chains by the number of carbons and summing precursor isotopologue values at a given time point to obtain natom equivalents of $^{13}C$ (Fig. 4c, Supplementary Data 1, Tables 8 and 9).

### Statistics and reproducibility

Both dual-labeling and lipid labeling experiments were performed independently and reproduced at least twice to ensure robustness of the results. A minimum of three (lipid labeling) and four (dual labeling) independent biological replicates were used. Chi-squared goodness of fit, 95% confidence intervals or k-means clustering were used as appropriate, for statistical analyses and were described within the figure legends and methods section.

### Reporting summary

Further information on research design is available in the Nature Portfolio Reporting Summary linked to this article.

### Data availability

Raw and pre-processed data for stable isotope labeled metabolomics using Arabidopsis roots and lipidomics using Camelina seeds is available at the NIH Common Fund's National Metabolomics Data Repository (NMDR) website, the Metabolomics Workbench, https://www.metabolomicsworkbench. org/, where they have been assigned study IDs ST002240 and ST002239 respectively[99]. The data for metabolomics and lipidomics experiments can be accessed directly via their project ID, PR001429 using the https://doi.org/10. 21228/M80X3B.

### Code availability

The tool *SIMPEL* is freely available for public use at https://github.com/ SIMPELmetabolism/SIMPEL, as an R package along with test data and source code. A tutorial is also included within GitHub with detailed instructions on SIMPEL usage with the test data. The tool is also available at the metabolomics workbench at https://www.metabolomicsworkbench. org/data/simpel_load.php, as a web tool.

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

## Acknowledgements
This work was supported by the National Institute of Health (NIH-U01 CA235508), National Science Foundation (NSF-IOS-1829365), Natural Sciences and Engineering Research Council of Canada (NSERC, RGPN-2018-05478), and the United States Department of Agriculture, National Institute of Food and Agriculture (2023-67017-39419) and Agricultural Research Service (USDA-ARS). Support for the acquisition of instruments used in this work were obtained using NSF (DBI-1827534). The authors would like to acknowledge the Subterranean Influences on Nitrogen and Carbon (SINC) Center, and the Proteomics and Mass Spectrometry Facility, and the Plant Growth Facility at the Donald Danforth Plant Science Center for facility and instrument use and assistance with instrument maintenance and plant growth. Special thanks to Ravi Kiran Garimella, Rahul Anand, and Jiahong Zhou for assistance in software design. Thanks also to Dr. Justin Renaud for his assistance with qExactive data acquisition and analysis.

## Author contributions
S.Ka., D.K.A. conceived and designed the experiments, wrote the initial manuscript, S.Ka., A.H.H., S.Ko., J.D.G. performed research and programming and contributed to manuscript writing, B.S.E. provided guidance on instrument setup and data acquisition, F.M., J.D.Y. contributed to experimental design, result interpretation and manuscript writing. All authors reviewed the manuscript.

## Competing interests
The authors declare no completing interests.
