## [Peer review file · Communications Biology]

Reviewers' comments:

Reviewer #1 (Remarks to the Author):

This paper developed a new metabolomic analysis by introducing dual-isotope labeling to identify new information in flux analysis. 2 examples are analyzed to prove the application of this method in both metabolomics and lipidomics. Overall this paper brings novel development into metabolomics analysis and the method maybe of interest to more fields. However, some questions are still needed to be clarified to improve the paper.

- 1) Dual-isotope labeling is cool, but also added difficulties to analysis if the resolution of MS is not high enough. I wonder if the authors can add comparisons between $^{13}\text{C}^{15}\text{N}$ dual-isotope labeling and 2 separate experiments with ^{13}C and ^{15}N single-isotope tracers?
- 2) It is pretty cool to see some difference between ^{13}C and ^{15}N fluxes in the first example. But overall the 2 fluxes seemed to have pretty similar trend, I wonder if the authors can add some more discussion about the meaning behind this? Or if it will be possible to use some genetical tools to prove the meaning of the difference in fluxes.
- 3) The lipidomics example is ^{13}C single isotope labeled. Maybe consider to put this example first.

Reviewer #2 (Remarks to the Author):

Kambhampati et al. developed a tool SIMPEL that quantifies isotope labeling results in high resolution metabolomics data, and incorporates global analysis and visualization function in a ready-to-use package. The utility of the tool is demonstrated with 2 examples. In the first example, analysis of a dual-labeled (^{13}C , ^{15}N) experiment shows more accurate flux estimation in Arabidopsis seedlings compared to a single-labeled (^{13}C) experiment. In the second example, analysis of ^{13}C labeled lipidomics data of developing pods results in a model of acyl editing mechanism in seed oil biosynthesis. The reviewer finds a few things that could be further clarified.

1. The manuscripts highlight the importance of a more efficient tool in analyzing isotope labeling metabolomics data, but does not mention much of existing tools with similar purpose (10.1038/s41467-022-31268-6; 10.1007/978-1-4939-9236-2_19; 10.1021/acs.analchem.7b02518; 10.1021/acs.analchem.5b03628; 10.1021/ac403384n). It will be helpful to further highlight the strength or unique features of SIMPEL.

2. The advantage of using dual-labeled (^{13}C , ^{15}N) over single-labeled (^{13}C) experiment is clearly demonstrated. Some existing tools have reported to process single-labeled data already. If the authors can demonstrate dual-labeled (^{13}C , ^{15}N) data is better than two separate single-labeled ^{13}C and ^{15}N data combined in flux calculation, SIMPEL being able to handle dual-labeled data would be more useful than other tools.

3. Distinguishing the incorporation of two ^{13}C groups ($\Delta m/z = 2.00671$) from a degree of fatty acid unsaturation ($\Delta m/z = 2.01565$) is important in ^{13}C lipidomics data analysis.

(1) What resolution is required to distinguish between PC(36:5)+ two ^{13}C groups and PC(36:4)? A user guide about HRMS setup before using the tool may be helpful to avoid data misannotation due to insufficiency mass spec resolution.

(2) It is impressive the demonstration study shows the analysis of ^{13}C enrichment within the fatty acid chain of PC and DAG molecular species (fig 2d). It is unclear whether such analysis is directly achieved in SIMPEL or is obtained through other manual analyses. It will be very useful if SIMPEL can perform tandem MS2 isotope labeling analysis.

4. As the author mentioned with cited ref 4 and 8, "PC may be a conduit for the direct shuttling of acyl chains from the chloroplast to the ER". What is new in the understanding of seed oil biosynthesis (or other processes) from this study compared to previously known knowledge? It may be difficult to extract all the important information from one figure for a general metabolomics background audience, as the seed oil biosynthesis process could be very complicated.

5. R Package installation and preliminary tests according to supplemental file 3 have been successful. Nonetheless, while examining the accompanying dataset or testing with our data, we find the error bar in the plot (similar to fig2a) is larger than that calculated from raw data.

Reviewer #3 (Remarks to the Author):

In this manuscript, the authors present a new tool for analyzing labeling data from metabolomics experiments that does not rely on an a priori model of a metabolism of interest. This work focuses on maximizing the strengths of high-resolution mass spectrometry to resolve labeling patterns measured using different atomic tracers (eg ^{13}C and ^{15}N). The authors have a history of making advanced approaches in processing labeling data accessible to a broader audience and this work is no exception. The manuscript is clearly written and the documentation for the R package is excellent.

This tool is likely to be very helpful to many researchers in the field and make improved, high resolution mass spec analysis and flux approaches more accessible.

Major:

It would be helpful to include the actual model file (with constraints/experimental settings, etc...) used for the INCA MFA that includes the dual ^{13}C and ^{15}N as a supplemental file or at least to indicate that it is to be made available upon request. I do appreciate that at least the reaction network with labeling structure is in the supplemental file 1

Minor:

Line 16: Shouldn't this read "...are evaluated..." since both fluxes and metabolic phenotypes are plural?

Supplemental notes: The definition of active and inactive pools could be made more precise. While it is true that often metabolically active pools are labeled and inactive pools remain unlabeled it is also true that a pool can be metabolically active yet only partially labeled, as appears to be the case with the calvin benson cycle that receive carbon from another source other than rubisco carboxylation. Perhaps a different term is needed or maybe just a clearer definition at the first mention, the authors clearly understand this effect as discussed later on as they point out the utility of using dual labels. Tom Sharkey has some recent work on this:

Reimport of carbon from cytosolic and vacuolar sugar pools into the Calvin–Benson cycle explains photosynthesis labeling anomalies
Y Xu, T Wieloch, JAM Kaste, Y Shachar-Hill... - Proceedings of the National Academy of Sciences, 2022

Dear Editor Zhu and Reviewers:

Here we provide a point-by-point response to the requests by the reviewers.

Reviewers' comments:

Reviewer #1 (Remarks to the Author):

This paper developed a new metabolomic analysis by introducing dual-isotope labeling to identify new information in flux analysis. 2 examples are analyzed to prove the application of this method in both metabolomics and lipidomics. Overall this paper brings novel development into metabolomics analysis and the method maybe of interest to more fields. However, some questions are still needed to be clarified to improve the paper.

- 1) Dual-isotope labeling is cool, but also added difficulties to analysis if the resolution of MS is not high enough. I wonder if the authors can add comparisons between $^{13}\text{C}^{15}\text{N}$ dual-isotope labeling and 2 separate experiments with ^{13}C and ^{15}N single-isotope tracers?

→The authors appreciate and agree with the reviewer's suggestion to provide a comparison of tracer experiments. To this end, the manuscript now includes a modified figure 1 with a comparison of three models (Fig 1c), that include a single, ^{13}C tracer, two parallel ^{13}C and ^{15}N tracer experiments, or the ^{13}C tracer data, ^{15}N tracer data and additional inclusion of a dual-labeled $^{13}\text{C}^{15}\text{N}$ tracer, which represents the information obtained from a single HRMS run, but modeled as nominal m/z that can be easily incorporated into the modeling software as further explained in Fig 1. The measurements that are included within the network for all three flux maps were provided as supplementary file 1.

- 2) It is pretty cool to see some difference between ^{13}C and ^{15}N fluxes in the first example. But overall the 2 fluxes seemed to have pretty similar trend, I wonder if the authors can add some more discussion about the meaning behind this? Or if it will be possible to use some genetical tools to prove the meaning of the difference in fluxes.

→Through the elucidation of multiple flux maps we have emphasized differences, and specifically that some fluxes are better determined, including those that would be difficult to establish through single isotope experiments because of a lack of carbon bond rearrangements. We did not perform a genetic manipulation of plants which would be an entire extensive study in and of itself and beyond the scope of the current tool; however, we hope that the additional emphasis in text, and additional models and maps that focus the utility of the tool make the distinctions more obvious for the reader.

- 3) The lipidomics example is ^{13}C single isotope labeled. Maybe consider to put this example first.

→While the lipidomics example is a single ^{13}C label, we are introducing a few novel ways to analyze data that hasn't been extensively considered in the past by isotope labeling studies due to the complexity in data. For example, use of MS2 data to obtain information on fatty acid labeling within lipids, use of accurate quantification data to calculate natom equivalents at both MS1 and MS2 levels to enable label comparison between metabolites of different pool sizes, etc. We realized that the flow of text would work better by first introducing the benefits of dual labels with flux analyses that people are generally familiar with, followed by new ways of analyzing labeled data with single labels, using functionality within SIMPEL so we decided to just try to better describe the experiments and smooth our text further. Hopefully this is an acceptable compromise for the reviewer.

Reviewer #2 (Remarks to the Author):

Kambhampati et al. developed a tool SIMPEL that quantifies isotope labeling results in high resolution metabolomics data, and incorporates global analysis and visualization function in a ready-to-use package. The utility of the tool is demonstrated with 2 examples. In the first example, analysis of a dual-labeled (^{13}C , ^{15}N) experiment shows more accurate flux estimation in Arabidopsis seedlings compared to a single-labeled (^{13}C) experiment. In the second example, analysis of ^{13}C labeled lipidomics data of

developing pods results in a model of acyl editing mechanism in seed oil biosynthesis. The reviewer finds a few things that could be further clarified.

1. The manuscripts highlight the importance of a more efficient tool in analyzing isotope labeling metabolomics data, but does not mention much of existing tools with similar purpose (10.1038/s41467-022-31268-6; 10.1007/978-1-4939-9236-2_19; 10.1021/acs.analchem.7b02518; 10.1021/acs.analchem.5b03628; 10.1021/ac403384n). It will be helpful to further highlight the strength or unique features of SIMPEL.

→The manuscript was originally written as a brief communication targeted for nature methods, with limits on word count as well as bibliography. The authors opted to take advantage of the manuscript transfer services that the journal offered, resulting in the shorter version being sent out for review. The re-submitted version now includes a full article with expanded introduction that much more fully cites relevant literature including important articles such as the ones mentioned by the reviewer, and highlighting the unique features of SIMPEL i.e., ability to handle dual labels, integrated natural abundance correction, global analyses such as k-means clustering, PCA plots and heat maps, etc.

2. The advantage of using dual-labeled (^{13}C , ^{15}N) over single-labeled (^{13}C) experiment is clearly demonstrated. Some existing tools have reported to process single-labeled data already. If the authors can demonstrate dual-labeled (^{13}C , ^{15}N) data is better than two separate single-labeled ^{13}C and ^{15}N data combined in flux calculation, SIMPEL being able to handle dual-labeled data would be more useful than other tools.

→The authors appreciate the reviewer's suggestion for comparison of dual-labeled (^{13}C , ^{15}N) vs two separate ^{13}C and ^{15}N labeled experiments. The revised manuscript now includes three-way comparison as described in the response to Rev #1. The paragraph starting in Page 3, Line 36 is also updated to reflect the additional data and the additional maps elucidate the increased rigor and statistical assurance of fluxes from a more complete set of data. Maximizing information from a single experiment has additional advantages in that it is economical and time-efficient, especially when tissue is limited which has also been emphasized.

3. Distinguishing the incorporation of two ^{13}C groups ($\Delta m/z = 2.00671$) from a degree of fatty acid unsaturation ($\Delta m/z = 2.01565$) is important in ^{13}C lipidomics data analysis.

- (1) What resolution is required to distinguish between PC(36:5)+ two ^{13}C groups and PC(36:4)? A user guide about HRMS setup before using the tool may be helpful to avoid data misannotation due to insufficiency mass spec resolution.

→The resolution required can be calculated using the formula $R = M / \Delta M$, where M is the molecular ion that needs to be resolved and ΔM is the difference in m/z between ions that is to be distinguished. In order to distinguish PC(36:5) ($M = 780.5543$) from $^{13}\text{C}_2$ labeled PC(36:4) ($\Delta M = 0.009$), a resolution of $\sim 87,000$ FWHM is required. Since resolution is inversely proportional to m/z and the instrument resolution is typically represented for an m/z of 200 Da, it is essential to set the required resolution within the method at $\sim 140,000$ to get $\sim 87,000$ at $m/z > 780$ Da.

The current experiment, however, used C8 chromatography where PC molecular species with different degrees of desaturation were chromatographically separated. The problem of the need for HRMS still exists, as the 18:2 ($M = 279.2324$) and $^{13}\text{C}_2$ of 18:3 ($\Delta M = 0.009$) acyl chain labeling within PC(36:5) still needs to be distinguished at MS2 level. This requires a resolution $> \sim 31,000$ FWHM. The authors agree that a guide to determining the required resolution for acquiring HRMS data will be useful for the readers. We included a supplemental note and a statement pointing the readers to the supplement (Page 12, Lines 10-12), to address this point.

- (2) It is impressive the demonstration study shows the analysis of ^{13}C enrichment within the fatty acid chain of PC and DAG molecular species (fig 2d). It is unclear whether such analysis is directly achieved in SIMPEL or is obtained through other manual analyses. It will be very useful if SIMPEL can perform tandem MS2 isotope labeling analysis.

→The MS2 analyses to assess ¹³C enrichment within the acyl chains of fatty acids is performed upon manual integration as pointed out in Page 13, Line 35 of the revised manuscript (Line 284-285 in the old submitted version). Automation of MS2 analyses is beyond the scope of current manuscript. The acquisition of MS2 data from labeled lipids (or any large molecules with >20 isotopologues) via iterative MS2 is a relatively new instrument capability (AcquireX by Thermo Scientific™) and is not routinely used. We are currently in the process of determining the best way to pre-process MS2 data collected in an iterative manner before this can be incorporated into SIMPEL. Our goal, however, for the future is to incorporate such analyses into the next release version of SIMPEL.

4. As the author mentioned with cited ref 4 and 8, “PC may be a conduit for the direct shuttling of acyl chains from the chloroplast to the ER”. What is new in the understanding of seed oil biosynthesis (or other processes) from this study compared to previously known knowledge? It may be difficult to extract all the important information from one figure for a general metabolomics background audience, as the seed oil biosynthesis process could be very complicated.

→The description that “PC may be a conduit for the direct shuttling of acyl chains from the chloroplast to the ER” comes from the observation that PCs (using both MS1 and MS2 data from this study) get labeled significantly faster than DAGs. The cited references are based on cell suspension cultures of Arabidopsis as model. Complementary experiments may be necessary using genetic tools and/or isolated chloroplasts to confirm this observation in oil seeds. The authors acknowledge that it is difficult to extract this important information. In the process of expanding the manuscript to full article, we provided more description in the context of our data along with a detailed discussion citing additional literature (See Page 9, line 10 onward).

5. R Package installation and preliminary tests according to supplemental file 3 have been successful. Nonetheless, while examining the accompanying dataset or testing with our data, we find the error bar in the plot (similar to fig2a) is larger than that calculated from raw data.

→The authors sincerely appreciate the reviewer’s time in testing out the software to provide feedback! As with any metabolomics data analyses tools, it is always challenging to identify noise from real metabolites and in this case isotopologues as well. It may require a few iterations by changing the mz and rt tolerances within the “get_table_object” function to accurately group the correct isotopologues. A statement was added within the methods section (Page 12, lines 41-44) to convey this point. In addition, following the information on the usage of function, outlined in the tutorial; supplemental file 3 (“?get_table_objects”) also includes helpful tips on each of the parameters available to users. Optimizing pre-processing parameters prior to generating a table for SIMPEL will also help in reducing the noise that contributes to errors in isotopologue grouping.

Reviewer #3 (Remarks to the Author):

In this manuscript, the authors present a new tool for analyzing labeling data from metabolomics experiments that does not rely on an a priori model of a metabolism of interest. This work focuses on maximizing the strengths of high-resolution mass spectrometry to resolve labeling patterns measured using different atomic tracers (eg ¹³C and ¹⁵N). The authors have a history of making advanced approaches in processing labeling data accessible to a broader audience and this work is no exception. The manuscript is clearly written and the documentation for the R package is excellent. This tool is likely to be very helpful to many researchers in the field and make improved, high resolution mass spec analysis and flux approaches more accessible.

Major:

It would be helpful to include the actual model file (with constraints/experimental settings, etc...) used for the INCA MFA that includes the dual ¹³C and ¹⁵N as a supplemental file or at least to indicate that it

is to be made available upon request. I do appreciate that at least the reaction network with labeling structure is in the supplemental file 1

→The authors appreciate the reviewer's suggestion. The MATLAB model files for all three experimental designs are now included as supplemental files 4, 5 and 6. Supplemental file 1 is also edited to reflect the revised flux models.

Minor:

Line 16: Shouldn't this read "...are evaluated..." since both fluxes and metabolic phenotypes are plural?

→Thank you, this statement is now modified as part of the revised introduction.

Supplemental notes: The definition of active and inactive pools could be made more precise. While it is true that often metabolically active pools are labeled and inactive pools remain unlabeled it is also true that a pool can be metabolically active yet only partially labeled, as appears to be the case with the calvin benson cycle that receive carbon from another source other than rubisco carboxylation. Perhaps a different term is needed or maybe just a clearer definition at the first mention, the authors clearly understand this effect as discussed later on as they point out the utility of using dual labels. Tom Sharkey has some recent work on this:

Reimport of carbon from cytosolic and vacuolar sugar pools into the Calvin–Benson cycle explains photosynthesis labeling anomalies

Y Xu, T Wieloch, JAM Kaste, Y Shachar-Hill... - Proceedings of the National Academy of Sciences, 2022

→The authors appreciate the reviewer's feedback on this. As suggested in Xu et al., 2022, the active pool descriptions may not always be easily modeled due to the contribution of weakly labeled or unlabeled active pools that are a consequence of experimental conditions. We are excited that our results suggest, using dual labels could help better estimate the active pools, at least in cases where more than one element is involved in contributing to the pool. The descriptions of active and inactive pools are now moved to the main article from the supplement and are expanded in the result section. The suggested literature is also cited.

REVIEWERS' COMMENTS:

Reviewer #1 (Remarks to the Author):

The authors addressed my previous comments and I have no other concerns.

Reviewer #2 (Remarks to the Author):

The reviewer finds all comments are addressed properly. Congrats on the wonderful new tool.

Reviewer #3 (Remarks to the Author):

The Authors have addressed my concerns and comments with the original submission and have appeared to also address what the other reviewers pointed out.